# Development and Characterization of Zn-ZnO Nanocomposites for Enhanced Biodegradable Material Properties

**DOI:** 10.3390/ma18050938

**Published:** 2025-02-21

**Authors:** Johngeon Shin, Jaewon Choi, Yong Whan Choi, Seongsoo Kim, Injoo Hwang

**Affiliations:** 1Department of Batteries Science and Engineering, Silla University, Busan 46958, Republic of Korea; jeshin@silla.ac.kr; 2Department of Mechanical Engineering, Silla University, Busan 46958, Republic of Korea; choijaewon0707@gmail.com (J.C.); ccs08745@silla.ac.kr (Y.W.C.); sskim@silla.ac.kr (S.K.)

**Keywords:** nanocomposite, mechanical properties, biodegradation, biocompatibility, bioimplant materials

## Abstract

Zinc has attracted significant attention as a versatile material with potential applications in various fields, particularly in biomedical engineering. Despite its desirable characteristics, such as biodegradability and biocompatibility, the inherently low mechanical strength of zinc has been a major limitation for its broader use in clinical applications. To address this issue and enhance its mechanical performance without compromising its biocompatibility, a novel composite material was developed by mixing zinc oxide (ZnO) with zinc (Zn). ZnO is widely recognized for its high chemical stability, non-toxicity, and antimicrobial properties, making it an excellent additive for biomedical materials. In this study, Zn-ZnO nanocomposites were fabricated by uniformly dispersing ZnO nanoparticles into molten zinc using an ultrasonic processor. The uniform distribution of ZnO nanoparticles within the zinc matrix was confirmed, and the resulting nanocomposites demonstrated remarkable improvements in mechanical properties. Specifically, the hardness and tensile strength of the Zn-ZnO nanocomposites were increased by approximately 90% and 160%, respectively, compared to pure zinc. To evaluate the biodegradation behavior of the materials, both pure zinc and Zn-ZnO nanocomposite samples were immersed in phosphate-buffered saline (PBS) at 37 °C, simulating physiological conditions. The degradation rate was assessed by measuring the weight loss of the material over time. The biodegradation rate of the Zn-ZnO nanocomposites was found to be nearly identical to that of pure zinc under identical conditions, indicating that the addition of ZnO did not adversely affect the degradability of the material. These findings suggest that Zn-ZnO nanocomposites offer a promising solution for biomedical applications by combining improved mechanical properties with maintained biodegradability and biocompatibility.

## 1. Introduction

Zinc is extensively studied as a biocompatible and biodegradable material due to its ability to be safely absorbed by the human body without adverse effects [1,2]. These attributes make zinc a highly attractive candidate for biomedical applications, particularly as an implant material in cardiovascular stents and orthopedic devices [3,4]. Among biodegradable metals, zinc stands out for its relatively slower degradation rate compared to magnesium (Mg), which often degrades too quickly for effective clinical use. The moderate degradation kinetics of zinc make it particularly suitable for healing vascular occlusions, as it provides sufficient time for tissue regeneration and healing before the implant is completely absorbed [5,6]. However, despite its favorable biodegradation and biocompatibility properties, zinc’s low mechanical strength limits its applicability in load-bearing and other high-stress biomedical environments.

To address this limitation, various strategies have been explored to enhance zinc’s mechanical properties. These strategies include alloying zinc with other metals and forming composite materials by incorporating ceramic particles [7,8]. Zinc-based alloys such as Zn-Mg, Zn-Fe, Zn-Al, and Zn-Ag are relatively easy to fabricate and exhibit significantly improved mechanical strength. For example, the yield strengths of Zn alloys with Mg, Al, and Ag additions are reported to be 150 MPa, 200 MPa, and 250 MPa, respectively, compared to 35 MPa for pure zinc [1]. These improvements in mechanical performance are essential for ensuring that the material can withstand the stresses encountered in biomedical applications. However, it remains essential to evaluate whether enhancing mechanical properties through alloying retains adequate biocompatibility, biodegradability, and appropriate degradation rates, as the introduction of alloying elements may adversely affect these critical properties.

In addition to alloying, the reinforcement of zinc with ceramic nanoparticles has emerged as an alternative strategy for improving mechanical performance. Nanoparticles such as silicon carbide (SiC), magnesium oxide (MgO), tungsten carbide (WC), aluminum oxide (Al_2_O_3_), and titanium dioxide (TiO_2_) have been incorporated into zinc matrices to create metal matrix composites (MMCs) with superior strength and hardness [9,10,11,12]. These composites offer the advantage of tailoring mechanical properties without altering zinc’s fundamental chemical composition. However, while nanoparticles like MgO, WC, and TiO_2_ are considered relatively non-toxic, comprehensive studies are needed to fully evaluate their biocompatibility, particularly in long-term biomedical applications.

Composite materials are specifically engineered to enhance the properties of the base matrix by dispersing secondary reinforcing materials. For zinc, which is naturally ductile, the incorporation of ceramic nanoparticles creates a particle-dispersion-reinforced composite that combines improved mechanical performance with the inherent biodegradability, biocompatibility, and ductility of the base material [13,14]. Composites can also integrate nanoparticles with inherent therapeutic properties, enabling their use in a wide range of biomedical applications, including drug delivery, antimicrobial therapy, tissue regeneration, and anti-inflammatory treatment [15]. The success of such composites depends heavily on the uniform distribution of nanoparticles within the matrix, which presents a significant challenge during fabrication.

Fabrication methods for metal matrix composites are typically categorized into solid-state and liquid-state processes [16,17]. Solid-state methods, such as sintering, involve combining the metal matrix and nanoparticles under high pressure and temperature in their solid forms [18,19]. These processes can produce highly homogeneous materials with excellent mechanical properties, but their complexity and high production costs limit their scalability for industrial and biomedical applications. In contrast, liquid-state processes, such as casting, are widely regarded for their simplicity and cost-effectiveness in mass production. However, achieving a uniform distribution of nanoparticles in liquid-state processes remains difficult due to issues such as poor wettability between the nanoparticles and the molten metal matrix, as well as the significant density differences between these materials [20,21].

In this study, ZnO nanoparticles were incorporated into a zinc matrix to enhance the mechanical properties of zinc while maintaining its biodegradability and biocompatibility. ZnO is a zinc-based material widely used in various industrial and scientific fields such as in transparent electrodes for semiconductor devices, gas sensors, and paints, due to its excellent chemical stability and functional versatility [22,23]. In biomedical contexts, ZnO is of particular interest because of its antifungal and antibacterial properties. However, concerns regarding its potential genotoxicity and cytotoxicity necessitate further investigation into its biological and toxicological effects [24].

The fabrication of Zn-ZnO composites poses challenges due to the poor wettability and significant density differences between Zn and ZnO, which hinder uniform mixing in the molten state. To address this, an ultrasonic processor was employed to achieve the uniform dispersion of ZnO nanoparticles within the molten zinc matrix. The microstructure, nanoparticle distribution, and chemical composition of the resulting composites were analyzed using field-emission scanning electron microscopy (FE-SEM) and energy-dispersive X-ray spectroscopy (EDS).

Mechanical properties, including tensile strength and hardness, were evaluated to quantify the improvement in performance achieved through nanoparticle reinforcement. Additionally, the biodegradability of the pure zinc and Zn-ZnO nanocomposites was assessed by immersing samples in phosphate-buffered saline (PBS) at 37 °C, simulating physiological conditions. Weight loss measurements were conducted weekly to monitor the degradation rates and ensure that the composite materials retained the favorable biodegradation characteristics of pure zinc.

Zinc exhibits a degradation rate comparable to the ideal rate required for biodegradable stents in vascular therapy. However, its relatively low mechanical strength has limited its application as a biodegradable stent material.

In this study, Zn-ZnO nanoparticles were incorporated into zinc to fabricate Zn-ZnO composites, aiming to enhance mechanical strength while preserving the degradation rate of zinc. The findings of this study demonstrate the potential of Zn-ZnO nanocomposites as a material that combines enhanced mechanical properties with the essential biocompatibility and biodegradability required for biomedical applications.

## 2. Materials and Methods

### 2.1. Fabrication of Zn-ZnO Nanocomposites

The schematic procedure for preparing Zn-ZnO nanocomposite specimens is shown in Figure 1. To fabricate the Zn-ZnO nanocomposite, ZnO nanoparticles were dispersed into molten Zn using an ultrasonic processor. In order to prevent oxidation during the process, a 99.9% Zn bar was employed as the base material for the molten Zn under an argon atmosphere. The procedure involved adding both Zn and ZnO nanoparticles (99.99% purity, 500 nm particle size, US Research Nanomaterials, Inc., Houston, TX, USA) into a crucible, where the Zn was completely melted at a temperature of 550 °C for a duration of 30 min. Prior to mixing, the weight percentages of the Zn (which included both the Zn rod and the Zn nanoparticles) and the ZnO nanoparticles were set to 1%, 3%, and 5%, respectively. After these materials were combined, the mixture was agitated for 5 min using a mechanical shaker to ensure the proper dispersion of the nanoparticles before they were introduced into the molten Zn. A detailed breakdown of the weight percentages of Zn and ZnO used in this experiment is provided in Table 1. Once the nanoparticles were prepared, they were carefully added to the molten Zn in the crucible, which was maintained at 550 °C. Then, the mixture was subjected to further agitation using an ultrasonic processor (SONICS, New Town, CT, USA, 750 Watts Ultrasonic Processor operating at 20 kHz) for 1 min. This ultrasonic treatment was employed to achieve a uniform dispersion of the ZnO nanoparticles within the molten Zn, ensuring a homogeneous nanocomposite material. For the preparation of tensile test specimens, the molten Zn-ZnO nanocomposites were poured into a preheated cast iron mold that had been set to a temperature of 200 °C. The specimens were allowed to cool to room temperature, solidifying into the desired form for mechanical testing. In the case of biodegradation experiments, the Zn-ZnO nanocomposite materials, once cooled in the crucible, were further processed by rolling and shaping them into thin sheets.

### 2.2. Characterization of Zn-ZnO Nanocomposites

To observe the microstructure of the Zn-ZnO nanocomposites, the surfaces of the specimens were prepared using standard polishing techniques, which included the application of sandpaper, followed by polishing with abrasive and polishing cloth. The surface morphology and elemental composition of the nanocomposites were analyzed using a field-emission scanning electron microscope (FE-SEM) (S-4800, Hitachi High-Tech Corporation, Tokyo, Japan) equipped with an energy-dispersive spectroscope (EDS). Vickers micro-hardness measurements were performed using an HMV-G Micro Vickers Hardness Tester. For each hardness test, a load of 1.96 N was applied for 3 s. The hardness of each specimen was measured 10 times to ensure reliable and reproducible results. Tensile tests were conducted using a SHIMADZU, Kyoto, Japan, AG-100KN Xplus universal testing machine to assess the tensile strength of the Zn-ZnO nanocomposites. The tests were performed at a constant strain rate of 10 mm/min. To ensure the consistency and accuracy of the measurements, ASTM E8M standard specimens were used, which feature a threaded grip shoulder with a 20 mm diameter and gauge dimensions of 12.5 mm in diameter and 50 mm in length [25].

### 2.3. Degradation Method

The biodegradation rate of the materials was systematically evaluated through immersion tests conducted in phosphate-buffered saline (PBS) solution maintained at a physiological temperature of 37 °C. The test specimens included pure Zn and Zn-ZnO nanocomposites containing 3 wt% and 5 wt% ZnO, fabricated as thin films. Each specimen was prepared with a controlled surface area of 150 mm^2^ and an initial weight of 1.30 g to ensure consistency across tests. The biodegradation process was monitored by measuring the weight of each specimen at regular intervals of 7 days. The weight loss over time was calculated as the difference between the measured weight of the specimen and its weight from the previous measurement. To maintain a consistent testing environment and to minimize the influence of external contaminants, the specimens were carefully cleaned with ethyl alcohol after each immersion cycle. This cleaning step was crucial to remove any residual degradation products or surface impurities that could interfere with subsequent measurements. Following the cleaning process, the specimens were immersed in fresh PBS solution to continue the biodegradation test under standardized conditions.

## 3. Results and Discussion

### 3.1. Morphology

The microstructure of the Zn-ZnO nanocomposites was examined using field-emission scanning electron microscopy (FE-SEM), as shown in Figure 2. The FE-SEM images clearly reveal the distribution of ZnO nanoparticles within the Zn matrix, highlighting their morphological characteristics. Figure 2a,b show the microstructures of nanocomposites containing 3 wt% and 5 wt% ZnO, respectively. The bright particles visible in these images correspond to ZnO nanoparticles, which are uniformly dispersed throughout the Zn matrix. Notably, Figure 2b demonstrates that as the proportion of ZnO increases during the mixing process, the amount of ZnO incorporated into the solid matrix also increases. However, higher ZnO concentrations lead to the formation of some nanoparticle clusters, indicating localized agglomeration. To provide a closer examination of these agglomerated regions, Figure 2c presents an enlarged view of the rectangular area highlighted in Figure 2b. This magnified image offers a detailed representation of ZnO distribution and clustering within the Zn matrix. To further characterize the elemental composition of these regions, energy-dispersive spectroscopy (EDS) analysis was performed at two specific points labeled as “1” and “2” in Figure 2c. The EDS analysis results, presented in Figure 2d, provide the elemental composition of the solid-state phases. At point 1, only Zn was detected, indicating a pure zinc matrix region. In contrast, both Zn and oxygen were detected at point 2, confirming the presence of ZnO in that area. To complement this analysis, an EDS line scan was conducted around the bright particles corresponding to point 2, and the results are included as an inset in Figure 2d. The line scan image uses light blue to represent Zn and red to represent oxygen. Within the bright particles, the oxygen signal peaks sharply, indicating the presence of ZnO, while outside these regions, the oxygen signal diminishes, leaving only Zn detectable. This detailed microstructural analysis demonstrates that ZnO nanoparticles were relatively homogeneously distributed throughout the Zn matrix in the solid state. As the initial ZnO content added before melting increased, a proportional increase in ZnO content within the solid matrix was observed. However, higher ZnO concentrations also led to the appearance of localized clustering, highlighting the balance required to achieve uniform dispersion in the composite.

### 3.2. Mechanical Properties of Zn–ZnO Nanocomposites

The Vickers microhardness values for pure Zn, Zn–1% ZnO, Zn–3% ZnO, and Zn–5% ZnO nanocomposites are presented in Table 2 and graphically illustrated in Figure 3. The average microhardness values measured for pure Zn, Zn–1% ZnO, Zn–3% ZnO, and Zn–5% ZnO were 42.3 HV, 46.0 HV, 60.0 HV, and 80.0 HV, respectively, with corresponding standard deviations of 2.2, 2.4, 4.0, and 4.2. These results indicate a substantial improvement in microhardness as the ZnO content increased within the Zn matrix. Specifically, the addition of 5% ZnO nanoparticles resulted in a nearly 90% enhancement in microhardness compared to pure Zn, demonstrating the significant strengthening effect achieved through nanoparticle incorporation.

However, it is noteworthy that the standard deviation also slightly increased with higher ZnO content, suggesting increased variability in the hardness measurements. This variability is attributed, at least in part, to the localized aggregation of ZnO nanoparticles, as observed in the microstructural analysis shown in Figure 2b,c. Such agglomeration can lead to the uneven dispersion of reinforcing particles within the matrix, creating localized regions with different mechanical properties. The primary mechanism underlying the observed hardness enhancement in Zn-ZnO nanocomposites is dispersion strengthening [13]. This phenomenon arises from the uniform distribution of ZnO nanoparticles within the Zn matrix, which act as obstacles to dislocation motion during deformation. By impeding dislocation movement, the nanoparticles effectively increase the hardness and strength of the composite material [26]. Additionally, the hard ceramic nature of ZnO contributes to this reinforcement by improving the load-bearing capacity of the composite. These findings highlight the critical role of nanoparticle content and dispersion quality in achieving optimal mechanical properties in Zn-based composites. While the addition of ZnO significantly enhances hardness, care must be taken to minimize nanoparticle clustering, as this can compromise the uniformity of the material’s properties.

Figure 4 presents the tensile stress–strain curves for pure Zn and Zn-ZnO nanocomposites containing 1 wt%, 3 wt%, and 5 wt% ZnO nanoparticles. The ultimate tensile strength (UTS) values for pure Zn, Zn–1% ZnO, Zn–3% ZnO, and Zn–5% ZnO were measured to be 32.1 MPa, 35.6 MPa, 60.6 MPa, and 84.3 MPa, respectively. These results correspond to strains at the UTS points of 0.21%, 0.24%, 0.46%, and 0.67%, respectively, highlighting a significant enhancement in both strength and ductility with increasing ZnO nanoparticle content.

A particularly notable characteristic of the Zn-ZnO nanocomposites is the simultaneous improvement in ultimate tensile strength and strain at the UTS points as more ZnO nanoparticles were incorporated into the Zn matrix. This dual enhancement suggests that the addition of ZnO not only reinforced the matrix but also contributed to a more effective load transfer mechanism within the composite structure, likely due to improved nanoparticle–matrix interaction.

However, while the UTS and strain at the UTS point improved, the overall strains at fracture exhibited a decline as the ZnO content increased. For Zn–1% ZnO, Zn–3% ZnO, and Zn–5% ZnO nanocomposites, the fracture strains were 1.52%, 1.71%, and 1.35%, representing decreases of 19.6%, 9.5%, and 28.6%, respectively, compared to the fracture strain of pure Zn. This reduction in ductility at fracture can be attributed to the inherent brittleness of the ceramic ZnO nanoparticles, which may act as stress concentrators, leading to earlier crack initiation under tensile loading [27].

### 3.3. Degradation of Zn–ZnO Nanocomposites

The in vitro biodegradation behavior of pure Zn and Zn–ZnO nanocomposites was evaluated by immersing specimens in a phosphate-buffered saline (PBS) solution maintained at 37 °C for a period of eight weeks. This test aimed to simulate the physiological conditions that these materials would encounter in biomedical applications. Figure 5 shows the physical changes observed in the samples before and after the biodegradation process, including their appearance during immersion in PBS. As shown in Figure 5b, a noticeable formation of white reaction products was observed on the surfaces of the metal specimens following the completion of the degradation test. These reaction products are indicative of the surface interactions between the material and the PBS solution, likely resulting from the dissolution of Zn and subsequent precipitation of Zn-containing compounds such as Zn phosphates or carbonates.

Figure 6 shows the changes in weight loss observed due to the chemical reaction between the test specimens and the PBS solution over time. The inset graph in Figure 6 represents the net weight loss measured on a weekly basis. Notably, during the first week of the biodegradation process, the specimens exhibited a relatively high and variable rate of weight loss. However, after the second week, the weight loss stabilized, showing a consistent rate over time. This behavior was similarly observed for both pure Zn and Zn–ZnO nanocomposites, indicating that the incorporation of ZnO nanoparticles at the concentrations used in this study did not significantly impact the overall biodegradation rate.

The data presented in Figure 3 and Figure 4 demonstrate that the addition of ZnO nanoparticles to the Zn matrix notably enhanced the mechanical strength of the material while maintaining the inherent biodegradation rate of pure Zn. This combination of increased mechanical performance and consistent degradation behavior positions the Zn–ZnO nanocomposite as a promising candidate for biomedical applications, offering both mechanical reliability and favorable biocompatibility.

## 4. Conclusions

In order to meet the requirements for advanced bioimplant materials, Zn–ZnO nanocomposite materials were synthesized and evaluated for their mechanical and biodegradation properties. The fabrication involved the incorporation of ZnO nanoparticles into molten Zn, facilitated by ultrasonic processing to achieve a uniform distribution throughout the Zn matrix. The resulting surface morphology analysis confirmed that ZnO nanoparticles were relatively evenly distributed, demonstrating good integration within the matrix. For the nanocomposite containing 5% ZnO, mechanical testing revealed substantial improvements in both hardness and tensile strength, showing enhancements of approximately 90% and 160%, respectively, compared to pure Zn. In addition to mechanical property improvements, the in vitro biodegradation study conducted in phosphate-buffered saline (PBS) solution at 37 °C revealed that the rate of weight loss over time was comparable between pure Zn and Zn–ZnO nanocomposites, indicating that the addition of ZnO did not adversely affect the biodegradation behavior. This result shows that ZnO not only serves to reinforce the mechanical properties of Zn but also maintains a degradation behavior similar to that of pure Zn. These findings suggest that the incorporation of ZnO nanoparticles into Zn matrices can lead to bioimplant materials with enhanced mechanical strength while preserving essential biocompatibility and biodegradation characteristics. Consequently, Zn–ZnO nanocomposites show strong potential as innovative and effective materials for bioimplant applications.

## Figures and Tables

**Figure 1 materials-18-00938-f001:**
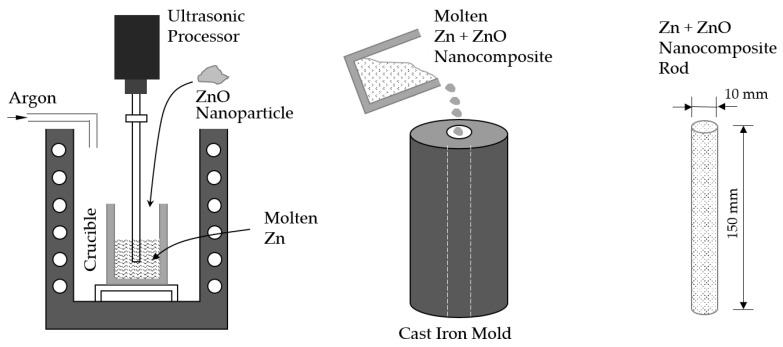
Schematic manufacturing procedure of Zn + ZnO nanocomposite rod.

**Figure 2 materials-18-00938-f002:**
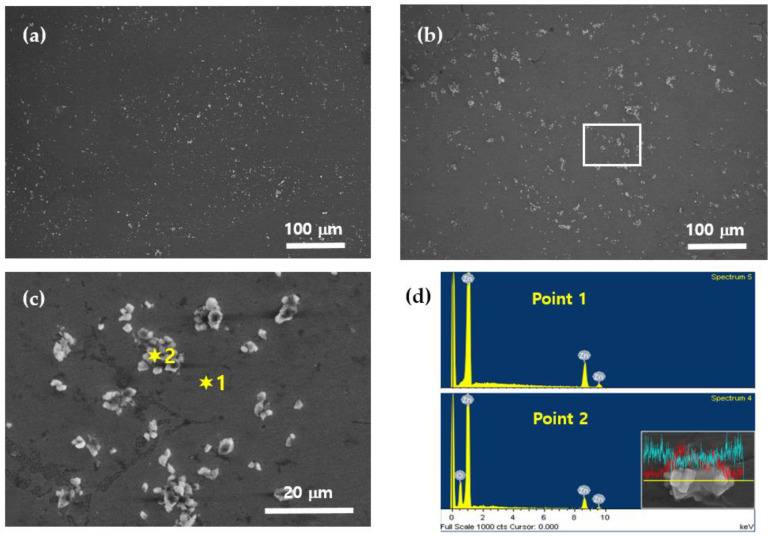
SEM and EDS images of Zn–ZnO nanocomposites: (**a**) Zn–3% ZnO; (**b**) Zn–5% ZnO; (**c**) enlarged image of the rectangular part in (**b**); (**d**) EDS analysis of Zn and ZnO nanoparticles.

**Figure 3 materials-18-00938-f003:**
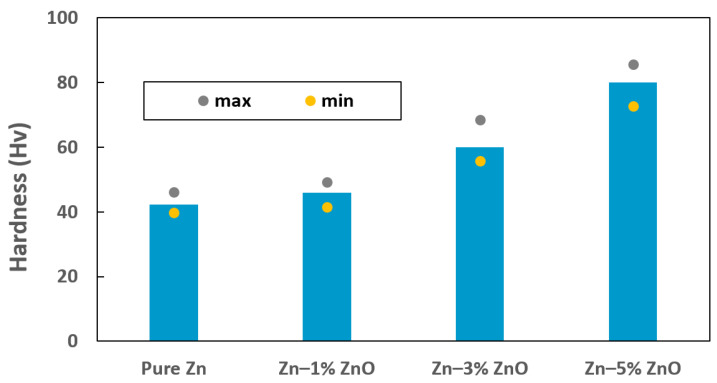
Vickers microhardness of pure Zn, Zn–1% ZnO, Zn–3% ZnO, and Zn–5% ZnO nanocomposites.

**Figure 4 materials-18-00938-f004:**
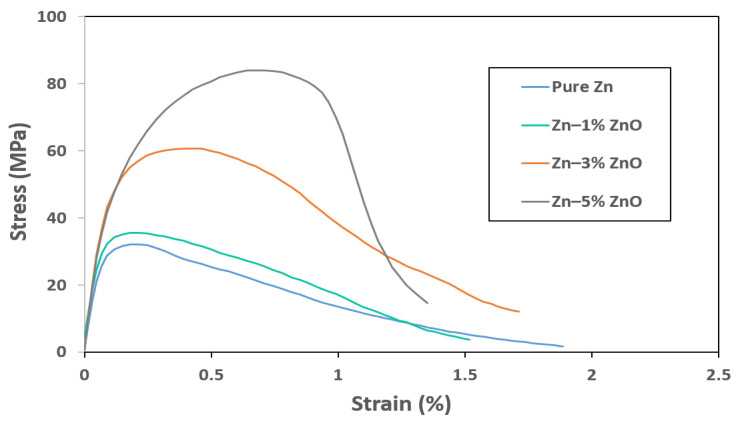
Tensile stress vs. strain curves for pure Zn, Zn–1% ZnO, Zn–3% ZnO, and Zn–5% ZnO nanocomposites.

**Figure 5 materials-18-00938-f005:**
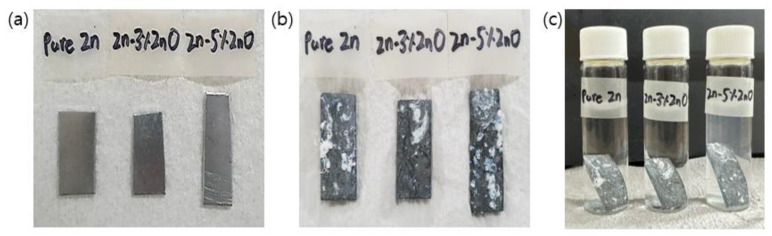
Images of Zn–ZnO nanocomposites before and after in vitro biodegradation test: (**a**) before test; (**b**) after 8 weeks; (**c**) placed in vials containing PBS.

**Figure 6 materials-18-00938-f006:**
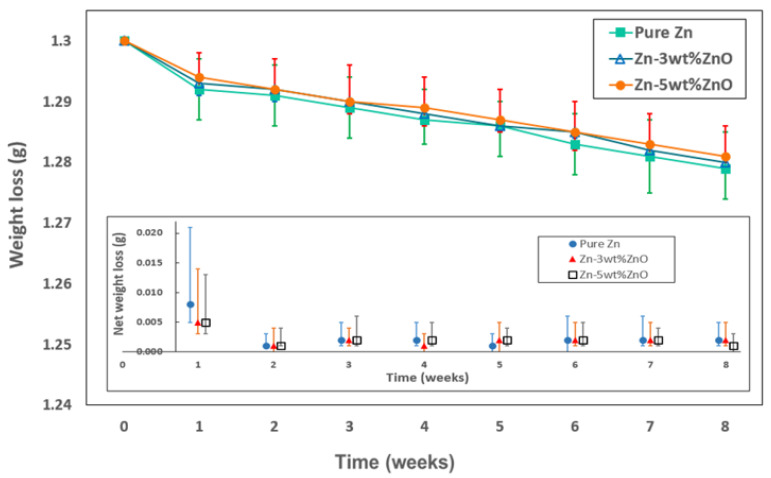
Weight loss due to biodegradation in PBS solution depending on immersion time. Inset plot shows net weight loss measured each week.

**Table 1 materials-18-00938-t001:** Composition of the Zn-ZnO nanocomposites.

	Zn Rod (g)	Zn Nanoparticle (g)	ZnO Nanoparticle (g)	Total Weight (g)
Pure Zn	130.0	0.0	0.0	130.0
Zn–1% ZnO	130.0	2.0	1.4	133.4
Zn–3% ZnO	130.0	2.0	4.2	136.2
Zn–5% ZnO	130.0	2.0	7.0	139.0

**Table 2 materials-18-00938-t002:** Microhardness (HV) of pure Zn, Zn–1% ZnO, Zn–3% ZnO, and Zn–5% ZnO.

	Pure Zn	Zn–1% ZnO	Zn–3% ZnO	Zn–5% ZnO
1	40.4	45.1	59.6	85.5
2	46.1	46.3	57.6	84.3
3	41.2	49.2	55.8	79.1
4	39.7	41.4	68.4	78.4
5	39.7	47.8	58.4	72.7
Average	42.3	46.0	60.0	80.0
Standard deviation	2.2	2.4	4.0	4.2

## Data Availability

Data are contained within the article.

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
