# Peer review of "Development and Characterization of Zn-ZnO Nanocomposites for Enhanced Biodegradable Material Properties"

_materials, 2025, doi:10.3390/ma18050938_

Round 1

Reviewer 1 Report (Previous Reviewer 1)

Comments and Suggestions for Authors

The article describes an innovative composite material consisting of a zinc (Zn) matrix reinforced with zinc oxide (ZnO) nanoparticles, which significantly improves the material's mechanical properties, without compromising its biodegradability and biocompatibility. Combining Zn and ZnO led to an increase in hardness and tensile strength of 90% and 160% compared to zinc alone. However, considering the potential biomedical applications of these materials, it is essential to conduct cytotoxicity tests to evaluate their biocompatibility. Tests on human cell lines such as fibroblasts or human endothelial cells (HUVEC) could provide crucial information to confirm the absence of toxic effects on cells and ensure the safety of the material for medical applications, such as bioabsorbable implants.

Author Response

Comments 1:[However, considering the potential biomedical applications of these materials, it is essential to conduct cytotoxicity tests to evaluate their biocompatibility. Tests on human cell lines such as fibroblasts or human endothelial cells (HUVEC) could provide crucial information to confirm the absence of toxic effects on cells and ensure the safety of the material for medical applications, such as bioabsorbable implants.]

Response 1:[Thank you for your insightful comment. We fully concur with your observation. The materials developed in this study are intended for eventual application in medical devices designed for implantation within the body. As such, future research will focus on evaluating the cytotoxicity and biocompatibility of the proposed materials to ensure their suitability for clinical use. Here we attach a summary of the acute toxicity test and the full test report, although not the cytotoxicity test results of Zn-ZnO nanocomposites.]

Reviewer 2 Report (New Reviewer)

Comments and Suggestions for Authors

The paper need major revisions.

1. The paper is too brief in terms of scientifi analysis and characterization of results.

2. Performa  a thorough review of literature and include the rationale to the objectives of your research at the end of the introduction.

2. Why Zn-ZnO was selected over other materials should be clearly justified in the introduction.

3. How was biodegardability analyzed without the biological reactions explained?

4. What are the nature of degaradation should be discussed.

5. The SEM images show no clear degradation.

6. The mechanical properties can be analysed after gradual degradation. If not, give necessary microscopic images after each interval.

7. The title is too general. What material, what composite? What resin?

8. Explain your material in sufficient details.

9. What kind of tensile test was done? which standard was followed?

10. What is teh statistical significance of the results?SD, CV?

11. Which standard for hardness measurement? Cite teh standard.

12. Where are the iages of the real samples? Too obscureimages.

13. Weigh loss can have other reasons. Why assume degradation? Supplement with more pictures or results.

Comments on the Quality of English Language

Minor revision

Author Response

Comments 1:[The paper is too brief in terms of scientifi analysis and characterization of results.]

Response 1:[ Thank you for your valuable feedback. We acknowledge your concern regarding the depth of scientific analysis and characterization of the results. To enhance the rigor of our study, we have expanded the discussion by providing a more detailed interpretation of the experimental results and incorporating additional characterization data where applicable. Specifically, we have:

  1. Expanded Data Analysis – We enhance the reliability and statistical significance of our findings, we have increased the number of specimens analyzed for physical properties.
  2. Additional Characterization – We provide a more comprehensive evaluation of the biocompatibility of the composites, we have incorporated the results of the acute toxicity test.

These revisions aim to improve the clarity and depth of our scientific analysis. We appreciate your insightful comments, which have helped us refine our manuscript, and we hope that the revised version adequately addresses your concerns.]

Comments 2:[Performa  a thorough review of literature and include the rationale to the objectives of your research at the end of the introduction.]

Response 2:[Thank you for your valuable insight. As you mentioned, we have elaborated on the rationale for selecting Zn-ZnO composites in the final part of the introduction section.]

Comments 3:[How was biodegardability analyzed without the biological reactions explained?]

Response 3:[Thank you for highlighting this critical aspect. PBS solutions provide a useful experimental environment to evaluate the early degradation behavior of biodegradable stents. In particular, it can be used to study the in vivo degradation rates and mechanisms of metal-based (Zn, Mg, Fe) and polymer-based (PLA, PLGA) stents. However, there are limitations to fully reproduce the actual in vivo environment, so further body fluid simulation and in vivo experiments may be required. In this study, as an initial study of the degradation behavior of Zn-ZnO composites, the degradation behavior was analyzed in PBS solution. We believe that analyzing the degradation behavior through biological reactions using animals will provide more accurate material properties than the current ones.]

Comments 4:[What are the nature of degaradation should be discussed.]

Response 4:[Thank you for your insightful question. This study represents an early-stage investigation of Zn-ZnO composites for biodegradable stent applications. In our analysis, we define degradation as the reaction of the specimen with the PBS solution, resulting in a loss of mass.]

Comments 5:[The SEM images show no clear degradation.]

Response 5:[The SEM image presented here illustrates the microstructure of the Zn-ZnO nanocomposite. It depicts the distribution of the reinforcing ZnO within the Zn matrix and provides insight into the extent of its agglomeration.]

Comments 6:[The mechanical properties can be analysed after gradual degradation. If not, give necessary microscopic images after each interval.]

Response 6:[Thank you for mentioning this important point. As you correctly pointed out, the mechanical properties can be analyzed after degradation. However, conducting mechanical property analyses at each interval for every specimen would require a large number of samples, which could constitute an entire study on its own. Given this, analyzing the mechanical strength of specimens at each degradation interval is beyond the scope of this study. We plan to investigate the mechanical strength of specimens after degradation in a future study.]

Comments 7:[The title is too general. What material, what composite? What resin?]

Response 7:[Thank you for your question. In this study, Zn-ZnO nanocomposites were fabricated by combining pure zinc as the matrix material with zinc oxide as the reinforcing material. This composite is entirely metal-based and does not contain any resin.]

Comments 8:[Explain your material in sufficient details.]

Response 8:[Thank you for your suggestion. This study represents an initial investigation into Zn-ZnO composites for biodegradable stent applications. Accordingly, the microstructure of the material was analyzed to assess the dispersion and potential agglomeration of zinc oxide nanoparticles, which serve as the reinforcing phase, within the zinc matrix. To enhance the mechanical properties of zinc, zinc oxide nanoparticles were incorporated, and various specimens were fabricated by adjusting the nanoparticle content. The mechanical strength of each specimen was evaluated through hardness and tensile tests. Additionally, the degradation behavior of pure zinc—known for its ideal biodegradation rate as a biodegradable stent material—was compared with that of Zn-ZnO composites using PBS solution.]

Comments 9:[What kind of tensile test was done? which standard was followed?]

Response 9:[Thank you for your question. The tensile test was conducted in accordance with ASTM E8M standards.]

Comments 10:[What is teh statistical significance of the results?SD, CV?

Response 10:[Thank you for your insightful question. We acknowledge the importance of assessing the statistical significance of our results to ensure their reliability. In this study, we analyzed the data by calculating the standard deviation to evaluate the variability within the measurements. To further enhance the statistical rigor of our study, we can incorporate additional statistical analyses, such as ANOVA or t-tests, to determine significant differences between groups. If deemed necessary, we will utilize these statistical assessments to strengthen the validity of our findings.]

Comments 11:[Which standard for hardness measurement? Cite teh standard.]

Response 11:[Thank you for your question. In this study, we measured the Vickers hardness using the HMV-G Micro Vickers Hardness Tester in accordance with the ASTM E384 standard to ensure precision and reliability in our hardness evaluations.]

Comments 12:[Where are the iages of the real samples? Too obscureimages.]

Response 12:[Thank you for your question. The samples were fabricated as plate tensile specimens using a casting mold. Specimens for SEM/EDS analysis and hardness testing were prepared by cutting the tensile specimens to the required size, followed by grinding and polishing. Meanwhile, the degradation specimens were cut into slightly different shapes while maintaining the same surface area to ensure consistency in degradation analysis.]

Comments 13:[Weigh loss can have other reasons. Why assume degradation? Supplement with more pictures or results.]

Response 13:[ Thank you for your question. There are various factors that contribute to weight loss. However, in this study, we focused on comparing the weight loss of Zn-ZnO composites with that of pure zinc, rather than investigating the specific causes of weight loss. As this study serves as a preliminary investigation for the application of biodegradable stent materials, it is crucial that the degradation rate of Zn-ZnO composites aligns with that of pure zinc. If the degradation rate differs significantly, the material may not be suitable for such applications. Fortunately, our results confirm that the degradation rate of Zn-ZnO composites is comparable to that of pure zinc. A detailed analysis of the causes of weight loss is beyond the scope of this study. However, we recognize its importance and plan to investigate the underlying mechanisms in future research.]

Reviewer 3 Report (New Reviewer)

Comments and Suggestions for Authors

The current manuscript aims to investigate the development and characterization of Zn-ZnO nanocomposites for enhanced biodegradable material properties. Although the topic is interesting in its scientific field, there are some issues that require the authors’ attention to improve the quality of this particular manuscript before further consideration for publication in a high-quality journal “Materials”.

Specific comments:

1.         According to the data presentation in Figure 6, there is no significant difference in the degradation rate between pure Zn and Zn-ZnO nanocomposites, implying no effect of ZnO addition. Why? Please justify.

2.         It is recommended to investigate other weight percentages of ZnO (e.g., 1% and 7%) to better understand the mechanical and degradation trends over a wider range.

3.         During the degradation process, the products may cause undesirable toxicity or irritating to surrounding tissues. Please examine the biocompatibility to ensure its future potential applications.

4.         Furthermore, whether the materials behave differently in distinct tissue environments? In addition to using PBS for degradation tests, relevant simulated physiological environments should be considered for degradability evaluations.

5.         Given that the current study emphasizes the use of ultrasonic processors to evenly disperse ZnO nanoparticles into molten zinc, the ultrasonic processing conditions such as frequency and power should be specified. Please improve.

6.         As mentioned in the Introduction section, composite materials are specifically engineered to enhance the properties of the base matrix by dispersing secondary reinforcing materials. Nevertheless, in my opinion, this important scientific claim is not supported by any appropriate documentation. If possible, please consider the inclusion of the following relevant report (DOI: 10.1016/j.cej.2022.134970) in the reference list to strengthen manuscript quality and attract more attention from broad readers.

Author Response

Comments 1:[According to the data presentation in Figure 6, there is no significant difference in the degradation rate between pure Zn and Zn-ZnO nanocomposites, implying no effect of ZnO addition. Why? Please justify.]

Response 1:[Thank you for highlighting this critical aspect. The Zn nanoparticles, serving as the matrix material, and ZnO nanoparticles, acting as the reinforcement, retain their intrinsic material properties even after being incorporated into the composite. Thus, the composite benefits from the ductility of the metal and the high strength of the ceramic, leading to enhanced mechanical properties. On the other hand, during the biodegradation process, Zn dissolves in the PBS solution, whereas ZnO remains intact. Consequently, in a composite containing a mixture of Zn and ZnO nanoparticles, as the Zn matrix gradually dissolves in PBS, the ZnO reinforcement nanoparticles naturally separate from the base material.

Our findings indicate that the weight percentage of ZnO has minimal influence on the degradation rate of the composite. This is because, as the Zn matrix degrades, the ZnO particles also detach from the composite. Consequently, the degradation rate of Zn-ZnO nanocomposites is comparable to that of pure zinc, the matrix material, and appears to remain constant degradation regardless of the ZnO weight ratio.]

Comments 2:[It is recommended to investigate other weight percentages of ZnO (e.g., 1% and 7%) to better understand the mechanical and degradation trends over a wider range.]

Response 2:[Thank you for your valuable suggestion. As per your suggestion, we incorporated Zn-1% ZnO specimens and conducted hardness and tensile tests. The Zn-1% ZnO sample exhibited an average hardness of 46.0 HV, representing a 9% increase compared to pure Zn. Similarly, the maximum tensile strength of Zn-1% ZnO reached 35.6 MPa, marking a 14% improvement over pure Zn. These results indicate that while Zn-1% ZnO demonstrates enhanced mechanical properties compared to pure Zn, the improvement is not substantial. We have implemented the above changes on pages 6 and 7 of the article and have also updated the corresponding tables and figures accordingly.

In addition, to increase the content of ZnO in Zn-ZnO composites (e.g. Zn-7% ZnO), we employed various approaches, including mechanical mixing, ultrasonic processing, and salt-assisted mixing. However, when the ZnO content exceeded 5 wt%, the nanoparticles tended to aggregate and settle at the bottom, resulting in uneven dispersion. Moreover, the significant surface tension of molten Zn posed challenges in preventing oxidation or burning of the ZnO nanoparticles during incorporation.

Moving forward, our research will focus on developing strategies to incorporate higher concentrations of ZnO nanoparticles while ensuring uniform dispersion. Additionally, we aim to determine the optimal zinc oxide content to maximize the mechanical properties of Zn-ZnO nanocomposites, as suggested.]

Comments 3:[During the degradation process, the products may cause undesirable toxicity or irritating to surrounding tissues. Please examine the biocompatibility to ensure its future potential applications.]

Response 3:[Thank you for your insightful comment. We fully concur with your observation. The materials developed in this study are intended for eventual application in medical devices designed for implantation within the body. As such, future research will focus on evaluating the biocompatibility and cytotoxicity of the proposed materials to ensure their suitability for clinical use.

Although the acute toxicity test results do not fully represent the overall biocompatibility of Zn-ZnO nanocomposites, we have provided a summary of the acute toxicity test along with the complete test report.]

Comments 4:[Furthermore, whether the materials behave differently in distinct tissue environments? In addition to using PBS for degradation tests, relevant simulated physiological environments should be considered for degradability evaluations.]

Response 4:[Thank you for mentioning this important point. To enhance the applicability of Zn-ZnO nanocomposites for implantable medical devices, we will investigate their degradation behavior in various simulated physiological environments. Additionally, where feasible, we will conduct preclinical animal studies to closely examine their degradation behavior in actual physiological conditions.]

Comments 5:[Given that the current study emphasizes the use of ultrasonic processors to evenly disperse ZnO nanoparticles into molten zinc, the ultrasonic processing conditions such as frequency and power should be specified. Please improve.]

Response 5:[Thank you for emphasizing this important aspect. As you correctly pointed out, optimizing the fabrication technique is essential for minimizing nanoparticle agglomeration in Zn-ZnO nanocomposites and achieving uniform dispersion. In this study, a 750 W ultrasonic processor was employed at 550°C to disperse the nanoparticles, utilizing 20 kHz vibrations for 1 minute. As you noted, factors such as the composite's melting temperature, frequency, wattage, and processing time are expected to affect nanoparticle dispersion. Future studies will systematically investigate variations in these parameters to identify the optimal conditions for Zn-ZnO nanocomposites.]

Comments 6:[As mentioned in the Introduction section, composite materials are specifically engineered to enhance the properties of the base matrix by dispersing secondary reinforcing materials. Nevertheless, in my opinion, this important scientific claim is not supported by any appropriate documentation. If possible, please consider the inclusion of the following relevant report (DOI: 10.1016/j.cej.2022.134970) in the reference list to strengthen manuscript quality and attract more attention from broad readers.]

Response 6:[Thank you for recommending an excellent review paper. We have incorporated the suggested paper into the introduction on page 2 of the manuscript and included it in the reference list.

If composites are fabricated using nanoparticles with inherent therapeutic properties, their potential applications are expected to expand significantly. We will carefully review the papers you recommended to identify nanoparticles suitable for zinc-based composites and enhance the quality of our research in biomedical applications.]

Reviewer 4 Report (New Reviewer)

Comments and Suggestions for Authors

This research paper authored by Shin etal. focused on the development of Zn-ZnO nanocomposites aimed at improving the mechanical properties of zinc while maintaining its biodegradability and biocompatibility for biomedical applications. ​ The authors successfully demonstrate that incorporating ZnO nanoparticles into a zinc matrix significantly enhances the material's hardness and tensile strength without adversely affecting its biodegradation. This research presents a novel method to address the low mechanical strength of zinc in biomedical applications by integrating ZnO nanoparticles, which are uniformly dispersed via a meticulously outlined ultrasonic fabrication process. Advanced characterization techniques such as FE-SEM, EDS, Vickers microhardness, and tensile testing offer substantial validation for the findings. The analysis integrates mechanical enhancements with biodegradation characteristics, maintaining biocompatibility without compromise. The results are presented clearly through effective figures and tables, which enhance the clarity and impact of the study. I have the following suggestions that could enhance the paper:

1. To advance the biomedical application of Zn-ZnO nanocomposites, it is crucial to conduct comprehensive cytotoxicity and genotoxicity studies to confirm their long-term safety.

2. Additionally, optimizing fabrication techniques to reduce nanoparticle clustering and achieve uniform dispersion is essential.

3. Long-term biodegradation tests should also be performed to better understand the material's behavior in physiological conditions, while scalability analyses are necessary to evaluate the feasibility of large-scale production.

Author Response

Comments 1:[To advance the biomedical application of Zn-ZnO nanocomposites, it is crucial to conduct comprehensive cytotoxicity and genotoxicity studies to confirm their long-term safety.]

Response 1:[Thank you for your insightful comment. We fully concur with your observation. The materials developed in this study are intended for eventual application in medical devices designed for implantation within the body. As such, future research will focus on evaluating the cytotoxicity and genotoxicity of the proposed materials to ensure their suitability for clinical use.

Here we attach a summary of the acute toxicity test and the full test report, although not the cytotoxicity test results of Zn-ZnO nanocomposites.]

Comments 2:[Additionally, optimizing fabrication techniques to reduce nanoparticle clustering and achieve uniform dispersion is essential.]

Response 2:[Thank you for highlighting this critical aspect. As you correctly pointed out, optimizing the fabrication technique is crucial to minimizing nanoparticle agglomeration in Zn-ZnO nanocomposites and achieving uniform dispersion. In this study, a 750 W ultrasonic processor was used at 550°C to disperse the nanoparticles, applying 20 kHz vibrations for 1 minute. As you noted, factors such as the composite's melting temperature, frequency, wattage, and processing time are expected to influence nanoparticle dispersion. Future studies will explore variations in these parameters to determine the optimal conditions for Zn-ZnO nanocomposites.]

Comments 3:[Long-term biodegradation tests should also be performed to better understand the material's behavior in physiological conditions, while scalability analyses are necessary to evaluate the feasibility of large-scale production.]

Response 3:[Thank you for your valuable suggestion. In this study, the results of the 8-week degradation test demonstrated that Zn-ZnO nanocomposites degraded at a consistent rate. We fully acknowledge the necessity of long-term biodegradation testing to gain a more comprehensive understanding of the material’s degradation behavior. Furthermore, we recognize the importance of developing efficient manufacturing methods to enhance the scalability and cost-effectiveness of these composites. In our future research, we plan to conduct the long-term biodegradation tests you mentioned and explore large-scale production techniques to improve the feasibility of Zn-ZnO nanocomposites as implantable materials for biomedical applications.]

Round 2

Reviewer 1 Report (Previous Reviewer 1)

Comments and Suggestions for Authors

The authors provided clarification regarding the future application of the developed materials. I am pleased to see that the research will focus on assessing cytotoxicity and biocompatibility, which are key to the clinical adoption of the materials. After reviewing the acute toxicity test summary and the full report, I believe the manuscript can be accepted for publication in its current form. I look forward to seeing the future developments of the research.

Reviewer 2 Report (New Reviewer)

Comments and Suggestions for Authors

May be accepted

Reviewer 3 Report (New Reviewer)

Comments and Suggestions for Authors

The revised version has adequately addressed most of the critiques raised by this reviewer and is now suitable for publication in "Materials".

This manuscript is a resubmission of an earlier submission. The following is a list of the peer review reports and author responses from that submission.

Round 1

Reviewer 1 Report

Comments and Suggestions for Authors

The article presents an interesting study on the use of Zn-ZnO nanocomposites to improve the mechanical properties of zinc, making it a more suitable material for biomedical applications. However, there are some areas where the article could be improved and deepened.

1. Improve characterization of nanocomposites: Although characterization experiments have been conducted to evaluate the mechanical properties and morphology of nanocomposites, other analyses, such as infrared spectroscopy, could be included to better understand the interactions between Zn and ZnO.

2. Additional biodegradability experiments: The article reports that the biodegradability of Zn-ZnO nanocomposites is similar to that of pure Zn, but it would be useful to include a more detailed characterization of the degradation kinetics over time and an evaluation of the degradation properties of the products of degradation.

3. Cytotoxicity and biocompatibility experiments: Since the nanocomposite could be used in biomedical applications, it would be important to conduct cytotoxicity and biocompatibility experiments to evaluate the effect of the materials on cells and tissues.

4. Manufacturing process optimization experiments: The article briefly mentions the nanocomposite manufacturing process, but does not discuss the optimization of this process in detail. Further experiments could be conducted to optimize the ratio of Zn to ZnO, as well as the fabrication conditions, in order to further improve the properties of the nanocomposites.

Overall, the article provides a solid foundation for further research on Zn-ZnO nanocomposites as materials for biomedical applications, but could benefit from more completeness in results and discussions.

Reviewer 2 Report

Comments and Suggestions for Authors

1-     In this study, the mechanical properties are directly related to the amount of ZnO. However, the authors did not test more samples with higher amounts of ZnO to demonstrate the optimum concentration of ZnO on the mechanical properties. It is suggested to add more samples and repeat these experiments.

2-     "As observed from the results depicted in Figure 3 and Figure 4, the incorporation of ZnO into the Zn matrix not only significantly increased the mechanical strength but also maintained the biodegradation rate of Zn. Generally, degradation rates are directly related to the mechanical properties of an implant. However, this trend was not observed in this study. Could the authors please explain the mechanism involved in retaining the biodegradation rate?"

3-     It is highly suggested to check the bioactivity and biocompatibility of the developed composites.

4-     What is the novelty of this manuscript and what benefits could be add in the scientific community by publishing this article? 

Comments on the Quality of English Language

The English language is fine and there are few grammars and typing errors.